# National Surveillance of Acute Flaccid Paralysis Cases in Senegal during 2017 Uncovers the Circulation of Enterovirus Species A, B and C

**DOI:** 10.3390/microorganisms10071296

**Published:** 2022-06-27

**Authors:** Ndack Ndiaye, Amary Fall, Ousmane Kébé, Davy Kiory, Hamet Dia, Malick Fall, Ndongo Dia, Amadou Alpha Sall, Martin Faye, Ousmane Faye

**Affiliations:** 1Virology Department, Institut Pasteur de Dakar, 36, Avenue Pasteur, BP 220, Dakar 12900, Senegal; ndack.ndiaye@pasteur.sn (N.N.); amary022@hotmail.com (A.F.); ousmane.kebe@pasteur.sn (O.K.); davy.kiori@pasteur.sn (D.K.); hametdia@outlook.fr (H.D.); ndongo.dia@pasteur.sn (N.D.); amadou.sall@pasteur.sn (A.A.S.); ousmane.faye@pasteur.sn (O.F.); 2Département de Biologie Animale, Faculté des Sciences et Techniques, Université Cheikh Anta DIOP de Dakar, BP 5005, Dakar 10700, Senegal; malickfal@yahoo.fr

**Keywords:** enterovirus, children, acute flaccid paralysis, Senegal

## Abstract

Polioviruses have been eliminated in many countries; however, the number of acute flaccid paralysis cases has not decreased. Non-polio enteroviruses are passively monitored as part of the polio surveillance program. Previous studies have shown that some enteroviruses do not grow in conventional cell lines used for the isolation of poliovirus according to the WHO guidelines. In order to evaluate the presence of enteroviruses, real-time RT-PCR was performed on Human Rhabdomyosarcoma (RD)-positive and RD-negative stool samples. A total of 310 stool samples, collected from children under the age of 15 years with acute flaccid paralysis in Senegal in 2017, were screened using cell culture and real-time RT-PCR methods. The selected isolates were further characterized using Sanger sequencing and a phylogenetic tree was inferred based on VP1 sequences. Out of the 310 stool samples tested, 89 were positive in real-time RT-PCR. A total of 40 partial VP1 sequences were obtained and the classification analysis showed that 3 (13%), 19 (82.6%), and 1 (4.4%) sequences from 23 RD-positive non-polio enterovirus isolates and 3 (17.6%), 7 (41.1%), and 7 (41.1%) sequences from 17 RD-negative stool samples belonged to the species EV-A, B, and C, respectively. Interestingly, the EV-B sequences from RD-negative stool samples were grouped into three separate phylogenetic clusters. Our data exhibited also a high prevalence of the EV-C species in RD-negative stool samples. An active country-wide surveillance program of non-polio enteroviruses based on direct RT-PCR coupled with sequencing could be important not only for the rapid identification of the involved emergence or re-emergence enteroviruses, but also for the assessment of AFP’s severity associated with non-polio enteroviruses detected in Senegal.

## 1. Introduction

Enteroviruses (EVs) are small (30-nm-diameter virions) non-enveloped viruses belonging to the picornaviridae family, *Enterovirus* genus. Evs include important human pathogens such as polioviruses, rhinoviruses, echoviruses, and coxsackieviruses. There are 15 species in the genus, and the types of viruses are the polioviruses that belong to the species C [1]. The enterovirus genome is a single-stranded positive-sense RNA molecule of approximately 7500 nucleotides, encompassed by a 5′ untranslated region (UTR) of approximately 750 nucleotides and a short 3′UTR of approximately 70–100 nucleotides. The large polyprotein translated from the single ORF encodes four structural proteins (VP1, VP2, VP3, and VP4) and seven non-structural proteins. The VP1 region has been correlated with the EV serotype and is used for the identification of EV types [2].

Established in 1988, the World Health Organization Global Polio Eradication Initiative (GPEI) contributed to an initial wealth of information on various enteroviruses, particularly polioviruses [3]. Poliovirus (PV) is the most well-known EV, causing acute flaccid paralysis (AFP) [4,5]. Global vaccination programs have significantly reduced the incidence of PV infections, and recently, some countries have been declared polio-free by the World Health Organization [6]. However, the number of AFP cases has not decreased in polio-free countries worldwide [7]. In Senegal, no new case of wild poliovirus was reported between 2010 and 2016 [8].

Beyond poliovirus, non-polio enteroviruses (NPEV) represent an important public health concern worldwide [9]. In Africa, only a few studies aimed at specifically identifying NPEV have been conducted and knowledge and resources for the specific detection of NPEV are limited. In addition, NPEV have been recently detected by RT-PCR from RD-L20B-negative stool samples collected from children with AFP [10,11]. In Senegal, an overall rate of NPEV of 20.3% was reported in children with AFP during 2017. This rate was significantly higher than those recorded during previous years [4,9,11].

Herein, we provide new insights into the molecular epidemiology of NPEV circulating in Senegalese children with AFP from January to December 2017, using cell culture-based isolation and direct RT-PCR methods for their detection.

## 2. Materials and Methods

### 2.1. Ethics Statement

As part of the GPEI, our study did not directly involve human participants, but included NPEV-positive supernatants from Human Rhabdomyosarcoma (RD) cells (RD-positive) and NPEV-negative stool samples (RD-negative) from children with AFP, collected as part of routine surveillance activities for polio in Senegal for public health purposes [12]. Our protocol has been approved by WHO and the national ethical committee at the Ministry of Health and Social Actions in Senegal considering all applicable national regulations governing the protection of human subjects. Clear oral consent was obtained from all patients or their parents or relatives.

### 2.2. Data Collection

From January to December 2017, a total of 310 stool samples were collected in the 14 medical regions in Senegal from 172 patients under 15 years old with AFP. The highest number of cases was reported in Dakar (Figure 1). Out of the 172 patients, two stool samples (24–48 h apart and within 14 days from the onset of paralysis) were collected from a total of 138 cases and one stool sample was obtained from 34 patients. All 310 samples were sent to the WHO-accredited Regional Reference Polio Laboratory in Senegal for processing according to the WHO standard procedures for poliovirus isolation [13].

### 2.3. Sample Preparation

Samples were treated using chloroform and stool suspensions were inoculated in both RD and human poliovirus receptor CD155 expressing recombinant murine (L20B) cell lines [14]. For samples showing no cytopathic effect (CPE) after 5 days, a second passage in the same cell line was done and cells were monitored for CPE during the next 5 days. Cell supernatants positive in RD cells but negative after 10 days in L20B cells were re-passaged in L20B cells and monitored for 5 days to exclude the possibility that they were polioviruses. Only cell supernatants producing CPE in RD cells and not in L20B cells were considered as NPEV. These RD-positive supernatants were then harvested and kept frozen (−20 °C) until typing.

### 2.4. RNA Extraction and Amplification

Viral RNA was extracted from 200 µL of RD-positive and RD-negative stool samples using the QIAamp viral RNA mini kit (Qiagen, Germany), according to the manufacturer’s recommendations. RNA extracts were screened for enteroviruses by rRT-PCR (rRT-PCR pan enterovirus) using the Light Mix^®^ Modular Enterovirus 500 kit (Roche-Ref 50-0656-96, TibMolBiol, Berlin, Germany) with the qScript™ XLT One-Step RT-PCR (Quanta Bio, Beverly, MA, USA), according to the manufacturer’s instructions.

For poliovirus intratypic differentiation assays (ITD), cell supernatants or extracted viral RNAs were screened using primer–probe mixture (contained in the ITD 5.1 kit; CDC, Atlanta, GA, USA). All experiments were performed using the CFX96TM Real-Time PCR system (Bio-Rad, Singapore).

For typing, if both stool samples from the same patient tested positive by RT-PCR, only one isolate was selected, while all RT-PCR-positive stool samples from patients with one specimen were further characterized. The selected RT-PCR-positive samples were characterized using a nested PCR method targeting the VP1 region, followed by Sanger sequencing, as previously described [15].

The association between viral load and the inability of isolating some species has been assessed using Ct values and data from the virus diversity analysis.

### 2.5. Sequencing and Phylogenetic Analysis

PCR products were purified and sequenced at Genewiz (Essex, UK) by the Sanger method, using the same forward and reverse primers that were used for PCR. The assembled sequences were curated with the GeneStudio software (GeneStudio ™ Pro, solvusoft, Microsoft, Redmond, DC, USA, version: 2.2.0.0), and the online Basic Local Alignment Search (BLAST) program (https://blast.ncbi.nlm.nih.gov/Blast.cgi; accessed on 20 August 2020) was used to assess the sequence homology with the previously available genomes. The genotyping was analyzed with the online RIVM program (http://www.rivm.nl/mpf/enterovirus/typingtool/, accessed on 20 August 2020). The generated sequences have been also analyzed using phylogenetic inference to confirm data from the classification of strains. Multiple alignments were performed using the MUSCLE method implemented in the MEGA 7.0 program [16]. The maximum likelihood (ML) tree was then constructed with partial VP1 sequences using FastTree v2.1.7 [17], with the best fit nucleotide substitution model to our sequence data. The ML tree was generated for 5000 replications and node support was assessed with the Shimodaira–Hasegawa test (SH-like) values. Topology was visualized by FigTree v.1.4.2 (http://tree.bio.ed.ac.uk/software/figtree/, accessed on 20 August 2020). Only SH-like values ≥ 0.8 were shown and a parechovirus strain was used as an outgroup.

### 2.6. Recombination Analysis

The presence of recombination events was assessed in the generated sequences using seven methods (RDP, GENECONV, MaxChi, BootScan, Chimaera, SiScan, and 3Seq) implemented in the Recombination Detection Program (RDP4.97) (http://web.cbio.uct.ac.za/~darren/rdp.html, accessed on 20 August 2020) [18]. The settings were kept at their default values and a recombination event was considered if it was detected by at least three different methods.

### 2.7. Statistical Analysis

To assess the possible influence of viral load on the cultivability of the strains, the mean Ct values have been compared between the RD-negative and the RD-positive stool samples for EV species, using the Student *t*-test. A *p*-value lower than 0.05 (*p* < 0.05) has been considered as significant.

## 3. Results

### 3.1. Isolation Data

In 2017, a total of 310 stool samples were collected from 172 children with AFP living in Senegal during a one-year surveillance period, including 20.3% (63/310) stool samples that exhibited CPE only in RD cells and classified as NPEV and 79.03% (245/310) of stool samples negative in both cell lines. The highest numbers of NPEV from AFP cases have been recorded between July and August with 10 and 13 isolates, respectively. These 63 RD-positive stool samples were from different cases. A total of 2 out of 310 stool samples (0.65%) were suspected for poliovirus (Figure 2). These two suspected polio samples originated from the same patient and were identified as Sabin-like 3 and a mixture of Sabin-like 1 with Sabin-like 3 for stool samples S1 and S2, respectively, using the intratypic differentiation test (IDT) for polioviruses.

### 3.2. Real Time RT-PCR Analysis

RNA extracts from all 310 stool samples were analyzed by RT-PCR. Overall, a total of 89 out of the 310 samples (28.7%) tested positive for enteroviruses by RT-PCR, including all 63 RD-positive stool samples and a total of 26 RD-negative stool samples. Overall, these 89 NPEV-positive stool samples were detected from 51.7% Senegalese children with AFP.

### 3.3. Virus Diversity

A total of 37 RD-positive stool samples and all 26 RD-negative stool samples were selected for typing. The VP1 protein was successfully amplified from 40 out of the 63 selected samples, including PCR products from 23 RD-positive stool samples and 17 RD-negative stool samples. The characterized VP1 sequences corresponded to 6 EV-A (15%), 26 EV-B (65%), and 8 EV-C (20%) species.

Out of the 23 RD-positive stool samples, a total of 3 (13%) belonged to the EV-A species, 19 (82.6%) to EV-B, and 1 (4.3%) to EV-C. Echoviruses were isolated from 14 cases (60.9%) and were the most prevalent EV-B species among the RD-positive stool samples, followed by coxsackievirus B5 with three cases (13.04%). Three (17.6%) out of the 17 RD-negative stool samples belonged to the EV-A species, seven (41.1%) to EV-B, and seven (41.1%) to EV-C. In addition, the most prevalent EV-C among the RD-negative stool samples included the EV-C99 and coxsackievirus A19 species (Table 1). The newly characterized sequences have been submitted to Genbank and their corresponding information is summarized in Appendix A (Appendix A).

The comparison between Ct values and data from the virus diversity analysis has shown that EV-A and EV-B species detected from the RD-negative stool samples exhibited Ct values significantly higher than those from the RD-positive stool samples belonging to the same species (*p* < 0.001). However, the Ct values of EV-C were relatively similar in both groups (*p* > 0.05).

### 3.4. Phylogenetic Analysis

A total of 37 VP1 sequences out of 40 had a length of approximately 375 bp and have been used for phylogenetic analyses, including NPEV isolates from 22 RD-positive and 15 RD-negative stool samples. In addition, the newly generated VP1 nucleotide sequences were aligned to 33 VP1 sequences retrieved from GenBank (https://www.ncbi.nlm.nih.gov/) (accessed on 18 May 2022). The phylogenetic tree was inferred using these 70 partial VP1 sequences representative of the current genetic diversity with the Kimura 2 parameters, with a Gamma distribution rate of 4 categories (Kimura 2 + G4) as the best nucleotide substitution model for our sequence data. The ML tree topology confirmed the genotyping data and showed that the EV-C sequences included only isolates from RD-negative stool samples that were grouped in the same cluster, while the EV-A cluster included sequences from both RD-positive and RD-negative stool samples. Interestingly, the EV-B strains were grouped into three separate clusters. Only sequences from RD-positive stool samples grouped into the cluster B1, while the clusters B2 and B3 included isolates from both RD-positive and RD-negative stool samples (Figure 3). In addition, there was no evidence for recombination detected in any NPEV sequences generated in our study.

## 4. Discussion

Enteroviruses represent a public health concern worldwide and the best-known species are polioviruses. However, little is known about the epidemiology of NPEV infections in Africa, where there exists a scarcity of data and a lack of diagnostic tools. Previous studies conducted in Senegal have shown the circulation of a large genetic diversity of human NPEV associated with acute flaccid paralysis (AFP), including mainly the EV-B species with a higher prevalence, followed by the EV-C and EV-D species [19,20]. Several factors could increase the risk of enterovirus transmission, particularly those contributing to the virus’ maintenance for a long period in environmental settings such as sewage [21].

In this study, we assessed the presence of NPEV from both RD-positive and RD-negative stool samples collected in 2017 from Senegalese children with AFP. Our data revealed a large diversity of NPEV circulating in Senegalese children with AFP, which belonged to the A, B, and C species. The detection rate of NPEV identified in our study was higher than those previously reported in other West African countries [19,20,22,23]. This difference can be associated with the fact that only RD-positive stool samples were characterized in these studies. Thus, our data provided more insights into the presence of NPEV in RD-negative stool samples since some NPEV species are not able to grow on RD cells [10,11].

The pan-enterovirus RT-PCR method exhibited a higher detection rate (28.7%) from stool samples than the cell culture-based isolation method (20.3%). These data provide an added value and reinforce the usefulness of direct RT-PCR as a suitable method for the detection of enteroviruses that could be applied in routine diagnosis activities, as previously described [24]. In addition, direct RT-PCR could be also combined with available room-temperature stable reagents and applied as a point-of-care diagnostic method, particularly in remote areas in Africa, where resources are still limited or inexistent [25].

In our study, several EV A, B, and C species were identified, including the CV-A4, CV-A14, and CV-A16 species that were previously reported as responsible for AFP [19,26,27]. Although these species have also been detected in healthy children [28], it could be important to pay attention to their possible involvement in the AFP syndrome in Senegalese children. In addition, the highest prevalence recorded in our study for the EV-B species (65%), which included mainly echoviruses, was similar to previous data reported in most of the AFP surveillance studies targeting NPEV [23,29,30,31,32]. Previously reported as responsible for neurological complications and AFP in children by the CDC and WHO, the circulation of echovirus 13 in Senegal needs to be further monitored [33,34,35].

The detection rate obtained in our study for the EV-C species from RD-positive stool samples was lower than those previously reported [22,23]. Interestingly, most of the EV-C sequences identified in our study originated from RD-negative stool samples and included mainly the EV-C99 and CV-A19 species. EV-C99 has been sporadically detected from both healthy children and children with AFP and among chimpanzees with AFP in DRC [19,23,28,29,36,37,38]. Thus, future studies relying on the assessment of risk factors such as direct or indirect contact with non-human primates, which lead to EV-C emergence in humans, could be further promoted [38].

Although the major proportion of EVs can be isolated on a wide variety of standard cell lines, the CV-A19 and CV-A22 strains detected in our study were not able to grow on the used cell lines [39,40,41]. The strains belonging to the coxsackievirus group A have been previously shown to grow poorly or not at all on cell lines currently approved by the WHO for use in the routine process for poliovirus isolation [42]. Thus, the difference in Ct values is consistent with the fact that the inability to grow in cell lines is related to the number of viral particles rather than to biological factors. However, the identification of a phylogenetic sub-clade that included only strains from RD-negative samples (species C) might suggest that the detectability can be type-dependent.

During the last five years, an increasing number of new non polio EV-C species have been identified [43] with a prevalence of around 50% in RD-negative stool samples from children with AFP [44,45,46]. These detection rates raise a question about the involvement of the cumulative presence of the NPEV-C species in the occurrence of AFP in children under 15 years old. Thus, it could be interesting to promote more studies with a focus on the specific implication of NPEV-C in the AFP syndrome, particularly in poliovirus-free countries.

The phylogenetic analysis exhibited no district-specific or month-specific cluster. Nevertheless, our data have shown that the molecular epidemiology of NPEV in children with AFP in Senegal during 2017 was similar to those previously described in West Africa and Asia [19,47,48]. However, the identification a sub-clade of EV-B including only RD-positive stool samples (B1 cluster) suggests that these strains could have specific biological properties that could be further characterized using a whole-genome sequencing approach.

Against this backdrop, it could be essential to establish specific NPEV surveillance programs based on the AFP syndrome worldwide, for the identification of the involved species, as most of them are ubiquitous.

## 5. Conclusions

Our study is noteworthy for reporting, for the first time, the molecular epidemiology of AFP-associated NPEV in Senegal, detected both in RD-positive and RD-negative stool samples. Our data confirm the high prevalence of NPEV related to AFP in Senegal, as previously described, and exhibited a high prevalence of EV-B species [28,49]. Our study showed also the presence of the EV-C species detected by direct RT-PCR, while they were not detected using cell culture-based isolation on suspected stool samples. However, combining both methods into a dual testing algorithm could be helpful in order to better describe the diversity of enteroviruses in any sample of interest. More studies are also needed to estimate the burden of the EV-C in Senegal and the different risk factors associated with the severity of the AFP syndrome. Implementation of the active surveillance of NPEV based on the existing national AFP surveillance program could play a pivotal role in the rapid identification of emerging or re-emerging NPEV of public health concern.

## Figures and Tables

**Figure 1 microorganisms-10-01296-f001:**
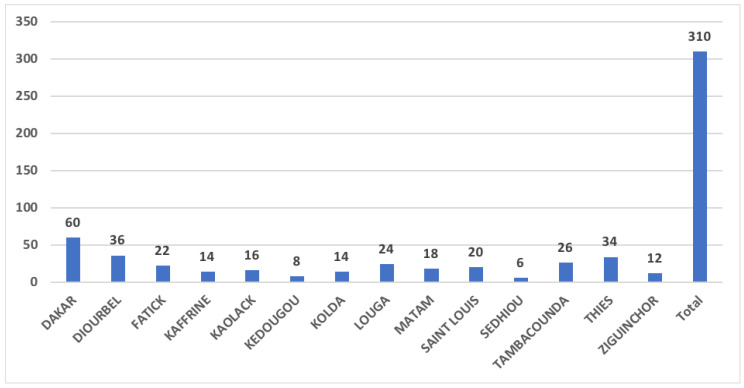
Geographical distribution of the number of stool samples from acute flaccid paralysis cases reported from January to December 2017 in Senegal. The diagram bars indicate the total number of reported AFP cases for each medical region.

**Figure 2 microorganisms-10-01296-f002:**
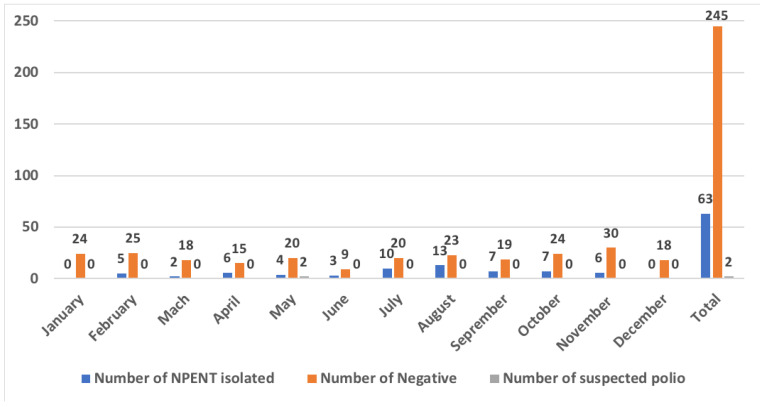
Temporal distribution of the selected stool samples in our study from January to December 2017 according to the cell culture results. The diagram bars highlighted in blue indicate the total number of isolated non-polio enteroviruses, while those color-coded in orange represent the number of samples that tested negative through the process used in our study. The samples suspected for polioviruses are highlighted in grey.

**Figure 3 microorganisms-10-01296-f003:**
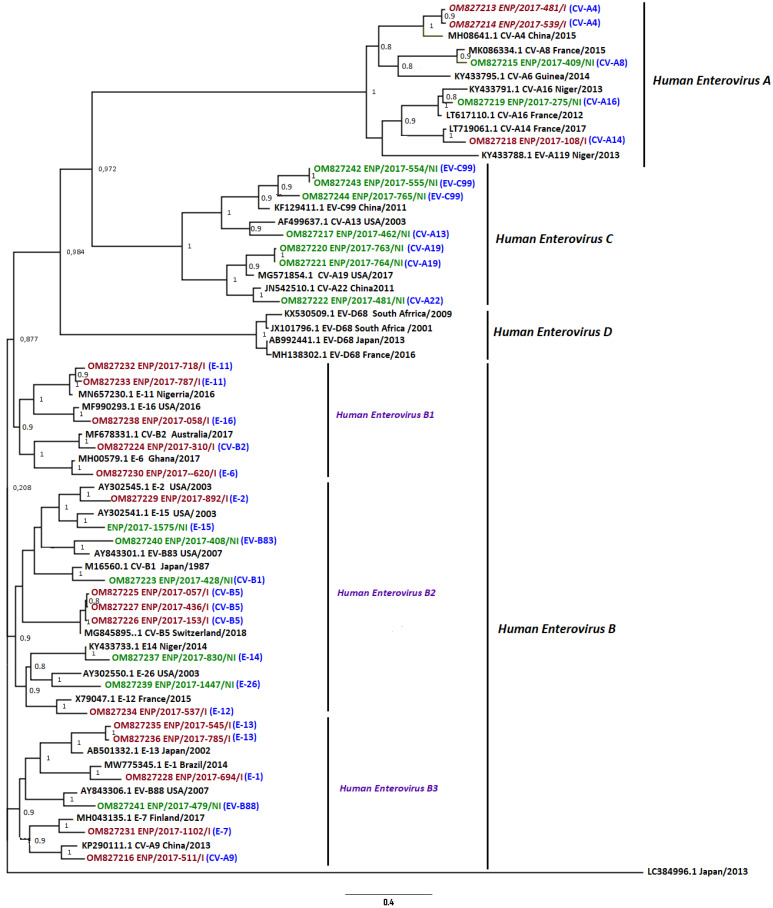
Maximum likelihood phylogenetic (ML) tree based on partial VP1 sequences (~3751 bp) identified from children with acute flaccid paralysis during 2017 in Senegal. The phylogenetic tree of the nucleotide sequences of the VP1 protein from 33 samples (18 RD-positive and 15 RD-negative stool samples). Multiple alignments and tree inference were performed using the MUSCLE and maximum likelihood methods, respectively, implemented in the MEGA 7.0 program. The method tree was generated for 5000 replications and nodes were supported by the SH-like values. SH-like values ≥ 0.8 are shown on the tree. The names in red indicate the strains from RD-positive stool samples; the names in green indicate the strains from RD-negative stool samples. The blue words refer to the abbreviation of type’s name and the purple words indicate the distinct clusters of enterovirus B species identified in our study. Sequences generated in this study belong to the A, B, and C enterovirus species.

**Table 1 microorganisms-10-01296-t001:** Sequencing results of non-polio enteroviruses and their classification.

Species	Type	Specimen Name	Total	Type	Specimen Name	Total
	Isolated (I)			Unisolated (NI)		
** *Enterovirus A* **	coxsackievirus A4	ENP/17-481 ENP/17-539	2	coxsackievirus A8	ENP/17-409	1
	coxsackievirus A14	ENP/17-108	1	coxsackievirus A16	ENP/17-275 ENP/17-408	2
** *Enterovirus B* **	coxsackievirus A9	ENP/17-511	1	coxsackievirus B1	ENP/17-428	1
	coxsackievirus B2	ENP/17-310	1	enterovirus B83	ENP/17-408	1
	coxsackievirus B5	ENP/17-057 ENP/17-153 ENP/17-436	3	enterovirus B88	ENP/17-479	1
	echovirus 1	ENP/17-694	1	echovirus 13	ENP/17-785	1
	echovirus 2	ENP/17-892 ENP/17-1125	2	echovirus 14	ENP/17-830	1
	echovirus 6	ENP/17-620	1	echovirus 15	ENP/17-1575	1
	echovirus 7	ENP/17-1102 ENP/17-408	2	echovirus 26	ENP/17-1447	1
	echovirus 11	ENP/17-718 ENP/17-787	2			
	echovirus 12	ENP/17-537	1			
	echovirus 13	ENP/17-545 ENP/17-785 ENP/17-1446	3			
	echovirus 16	ENP/17-058	1			
	echovirus 33	ENP/17-1409	1			
** *Enterovirus C* **	coxsackievirus A21	ENP/17-832	1	coxsackievirus A13	ENP/17-462	1
				coxsackievirus A19	ENP/17-763 ENP/17-764	2
				coxsackievirus A22	ENP/17-481	1
				enterovirus C99	ENP/17-554 ENP/17-555 ENP/17-765	3
**Total**			23			17

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
