# Peer review of "National Surveillance of Acute Flaccid Paralysis Cases in Senegal during 2017 Uncovers the Circulation of Enterovirus Species A, B and C"

_microorganisms, 2022, doi:10.3390/microorganisms10071296_

Round 1

Reviewer 1 Report

The article by Ndiaye et al. reports data that are highly relevant within the global surveillance of non-polio enterovirus circulation and of acute flaccid paralysis. The study is well performed, and this new version of the manuscript is significantly improved respect to the previous one. The manuscript is much clearer and better focused. I think this version is suitable for publication after some minor issues are fixes.

- Line 20. Change to “A total of 310 stool samples collected from children under the age of 15 years with acute flaccid paralysis”

- Line 21. It is not entirely clear which are the 2 methods. They should probably be specified in the text.

- Line 30. The article did non to next-gen sequencing, and it is weird that the abstract contains conclusions about it. I would replace “next-generation sequencing” with “sequencing”.

- Lines 52-3. Change to ‘’ In Senegal, no new cases of wild poliovirus have been reported between 2010 and 2016”.

- Lines 55-6. Change to “In Africa, only a few studies aimed at specifically identifying NPEV have been conducted”

- Line 60. A reference or some numbers should be provided to support this statement.

- Line 63. Remove “large”.

- Lines 76-78. Change to “Out of the 172 patients, two stool samples (24–48 hours apart and within 14 days from the onset of paralysis) were collected from a total of 138 cases and one stool sample was obtained from 34 patients”.

- Lines 119 and 203. MUSCLE should be written in capital letters.

- Lines 120-121. Change to “The maximum likelihood (ML) tree was then constructed with partial VP1 sequences using FastTree v2.1.7”.

- Line 123. Change to “and node support was assessed with the Shimodaira-Hasegawa test”.

- Line 150. Shouldn’t this also be included in the M&M section?

- Figure 2. I would specify in the figure caption that this refers only to cell-culture results.

- Line 161. Replace “isolated” with “detected”.

- Line 189. Typo: “Kimura 2 parameters”.

- Line 192. Shouldn’t this be “RD-negative”?

- Lines 208-9. This sentence should be removed from the figure caption.

- Line 249. Change to “from both healthy children and children with AFP”

- Line 255. Shouldn’t this be “strains” (not species)?

- Lines 257-264. Actually, the difference in Ct values is consistent with the fact that the inability of growing in cell-lines is related to the number of viral particles rather than to biological factors. On the other side, the fact that you had 1 clade in the tree that included only strains from RD-negative samples (species C) might suggest that the detectability can be type-dependent. This part should be adjusted.

- Lines 265. Change to “During the last five years”.

- Line 275-6. What do you mean with “isolation phenotype”? This sentence is not clear.

- Appendix A should be referred to in the text. 

Author Response

Open Review

English language and style

( ) Extensive editing of English language and style required
( ) Moderate English changes required
(x) English language and style are fine/minor spell check required
( ) I don't feel qualified to judge about the English language and style

Yes

Can be improved

Must be improved

Not applicable

Does the introduction provide sufficient background and include all relevant references?

(x)

( )

( )

( )

Are all the cited references relevant to the research?

(x)

( )

( )

( )

Is the research design appropriate?

(x)

( )

( )

( )

Are the methods adequately described?

(x)

( )

( )

( )

Are the results clearly presented?

( )

(x)

( )

( )

Are the conclusions supported by the results?

( )

(x)

( )

( )

Comments and Suggestions for Authors

The article by Ndiaye et al. reports data that are highly relevant within the global surveillance of non-polio enterovirus circulation and of acute flaccid paralysis. The study is well performed, and this new version of the manuscript is significantly improved respect to the previous one. The manuscript is much clearer and better focused. I think this version is suitable for publication after some minor issues are fixes.

- Line 20. Change to “A total of 310 stool samples collected from children under the age of 15 years with acute flaccid paralysis”

Response: This sentence has been revised according to the reviewer’s comments.

- Line 21. It is not entirely clear which are the 2 methods. They should probably be specified in the text.

Response: This sentence have been corrected in the revised manuscript.

- Line 30. The article did non to next-gen sequencing, and it is weird that the abstract contains conclusions about it. I would replace “next-generation sequencing” with “sequencing”.

Response: This sentence has been revised according to the reviewer’s comments.

- Lines 52-3. Change to ‘’ In Senegal, no new cases of wild poliovirus have been reported between 2010 and 2016”.

Response: This sentence has been revised according to the reviewer’s comments.

- Lines 55-6. Change to “In Africa, only a few studies aimed at specifically identifying NPEV have been conducted”

Response: This sentence has been revised according to the reviewer’s comments.

- Line 60. A reference or some numbers should be provided to support this statement.

Response: corresponding references have been added in the revised manuscript.

- Line 63. Remove “large”.

Response: This word has been removed added in the revised manuscript.

- Lines 76-78. Change to “Out of the 172 patients, two stool samples (24–48 hours apart and within 14 days from the onset of paralysis) were collected from a total of 138 cases and one stool sample was obtained from 34 patients”.

Response: This sentence has been revised according to the reviewer’s comments.

- Lines 119 and 203. MUSCLE should be written in capital letters.

Response: This word has been edited in the revised manuscript.

- Lines 120-121. Change to “The maximum likelihood (ML) tree was then constructed with partial VP1 sequences using FastTree v2.1.7”.

Response: This sentence has been revised according to the reviewer’s comments.

- Line 123. Change to “and node support was assessed with the Shimodaira-Hasegawa test”.

Response: This sentence has been revised according to the reviewer’s comments.

- Line 150. Shouldn’t this also be included in the M&M section?

Response: This remark has been added in the Materials and Methods section.

- Figure 2. I would specify in the figure caption that this refers only to cell-culture results.

Response: This remark has been corrected in the revised manuscript.

- Line 161. Replace “isolated” with “detected”.

Response: Isolated word have been replaced by detected in the revised manuscript.

- Line 189. Typo: “Kimura 2 parameters”.

Response: These typos have been corrected in the revised manuscript.

- Line 192. Shouldn’t this be “RD-negative”?

Response: This sentence has been edited in the revised manuscript

- Lines 208-9. This sentence should be removed from the figure caption.

Response: This sentence has been removed according to the reviewer’s comments.

- Line 249. Change to “from both healthy children and children with AFP”

Response: This sentence has been revised according to the reviewer’s comments.

- Line 255. Shouldn’t this be “strains” (not species)?

Response: This remark has been corrected in the revised manuscript.

- Lines 257-264. Actually, the difference in Ct values is consistent with the fact that the inability of growing in cell-lines is related to the number of viral particles rather than to biological factors. On the other side, the fact that you had 1 clade in the tree that included only strains from RD-negative samples (species C) might suggest that the detectability can be type-dependent. This part should be adjusted.

Response: This sentence has been revised according to the reviewer’s comments.

- Lines 265. Change to “During the last five years”.

Response: This sentence has been revised according to the reviewer’s comments.

- Line 275-6. What do you mean with “isolation phenotype”? This sentence is not clear.

Response: This sentence has been modified in the revised manuscript.

- Appendix A should be referred to in the text

Response: This remark has been corrected in the revised manuscript.

Reviewer 2 Report

Ndiaye et al

The manuscript is exceptionally well written and will make an important contribution to the literature on nonpolio paralysis when published.

Potential corrections:

The abstract refers to RD without defining it.  Also RD is not defined at first use (line 67) but is subsequently defined on line 87.  

Line 86: Chloroform does not need to be capitalized

Line 106: Nested does not need to be capitalized

Throughout the manuscript:  virus names are not ordinarily capitalized in most journals

Author Response

Open Review

English language and style

( ) Extensive editing of English language and style required
( ) Moderate English changes required
(x) English language and style are fine/minor spell check required
( ) I don't feel qualified to judge about the English language and style

Yes

Can be improved

Must be improved

Not applicable

Does the introduction provide sufficient background and include all relevant references?

(x)

( )

( )

( )

Are all the cited references relevant to the research?

(x)

( )

( )

( )

Is the research design appropriate?

(x)

( )

( )

( )

Are the methods adequately described?

(x)

( )

( )

( )

Are the results clearly presented?

(x)

( )

( )

( )

Are the conclusions supported by the results?

(x)

( )

( )

( )

Comments and Suggestions for Authors

Ndiaye et al

The manuscript is exceptionally well written and will make an important contribution to the literature on nonpolio paralysis when published.

Potential corrections:

The abstract refers to RD without defining it.  Also RD is not defined at first use (line 67) but is subsequently defined on line 87. 

Response: This remark has been corrected in the revised manuscript.

Line 86: Chloroform does not need to be capitalized

Response: This remark has been corrected in the revised manuscript.

Line 106: Nested does not need to be capitalized

Response: This remark has been corrected in the revised manuscript.

Throughout the manuscript:  virus names are not ordinarily capitalized in most journals

Response: This remark has been corrected in the revised manuscript.

This manuscript is a resubmission of an earlier submission. The following is a list of the peer review reports and author responses from that submission.

Round 1

Reviewer 1 Report

The authors described on active country-wide surveillance of non-polio enteroviruses using virus isolation and direct RT-PCR method.   They suggested that a wide range of enteroviruses apart from polioviruses are causative agents of acute flaccid paralysis cases in Senegal and the direct RT-PCR method is useful especially to detect EV-C strains.   As the authors described that little is known about the epidemiology of non- polio enteroviruses infection in Africa where there exists a scarcity of data and a lack of diagnostic tools, such an approach is important.   My comments are as follows.

1.(Line 42) “the type species of the genus is species C” is correct?

2.(Line 78) The authors described that 138 cases had two stool specimens.   Did you have any cases from whom the identical enterovirus was detected from both specimens?   If so, how did you use the data in this manuscript?

3.(Lines 86 and 87) Is there any difference between “sample” and “specimen”?

4.(Lines 88, 89 and 185) RD-L20B cell culture means RD cell culture and L20B cell culture?

  1. (Line 124) Two were suspected for poliovirus. Were they finally identified as poliovirus?  If so, which serotype?

  1. (Lines 136-137) How did you select these 26 RD-positive supernatants and RD-negative and RT-PCR positive specimens for typing? Why didn’t you analyzed all positive cases?

7.(Lines 142-143)   I recommend putting (Table 1) at the end of the sentence “Echoviruses were isolated … (15.8%).

8.(Line 146) “Tableau” may be French.

9.(Line 149) A total of 37?

  1. (Line 150) 13 should be 18?

11.(Figure 3) There are several questions related to Figure 3.  

  1. Did you register the sequences you analyzed? If so, please show the GenBank number.
  2. Why you did not include 37 cases shown in Table 1?
  3. Information of serotype should be included.
  4. Explanation for brown triangle and green circle should be included.
  5. Probably, 19 brown triangles indicate 19 RD-positive supernatants. According to Table 1, there are 3 enterovirus A, 15 enterovirus B, and 1 enterovirus C. However, there are 2 enterovirus A, 17 enterovirus B and 0 enterovirus C in Figure 3.   These data suggest that there is discordance of the results between RIVM program and phylogenetic analysis.    I guess one reason for such a discordance is that partial VP1 sequence were too short (~375bps) and regrettably I think the methodology in this study is not appropriate.

Author Response

Responses to Reviewer(s)' Comments:

Reviewer 1

Open Review

English language and style

( ) Extensive editing of English language and style required
( ) Moderate English changes required
( ) English language and style are fine/minor spell check required
(x) I don't feel qualified to judge about the English language and style

Yes

Can be improved

Must be improved

Not applicable

Does the introduction provide sufficient background and include all relevant references?

(x)

( )

( )

( )

Is the research design appropriate?

(x)

( )

( )

( )

Are the methods adequately described?

(x)

( )

( )

( )

Are the results clearly presented?

(x)

( )

( )

( )

Are the conclusions supported by the results?

( )

( )

( )

(x)

Comments and Suggestions for Authors

The authors described on active country-wide surveillance of non-polio enteroviruses using virus isolation and direct RT-PCR method.   They suggested that a wide range of enteroviruses apart from polioviruses are causative agents of acute flaccid paralysis cases in Senegal and the direct RT-PCR method is useful especially to detect EV-C strains.   As the authors described that little is known about the epidemiology of non- polio enteroviruses infection in Africa where there exists a scarcity of data and a lack of diagnostic tools, such an approach is important.   My comments are as follows.

1.(Line 42) “the type species of the genus is species C” is correct?

Response: This sentence has been rephrased in the revised manuscript.

2.(Line 78) The authors described that 138 cases had two stool specimens.   Did you have any cases from whom the identical enterovirus was detected from both specimens?   If so, how did you use the data in this manuscript?

Response: No, we didn’t have this case in our study.

3.(Lines 86 and 87) Is there any difference between “sample” and “specimen”?

Response: The expression “specimen” has been standardized to “sample” in the in the revised manuscript.

  1. (Lines 88, 89 and 185) RD-L20B cell culture means RD cell culture and L20B cell culture?

 Response: The expression “RD-L20B cell culture” have been modified to “RD-L20B-cell-culture” in the in the revised manuscript.

  1. (Line 124) Two were suspected for poliovirus. Were they finally identified as poliovirus?  If so, which serotype?

 Response: Details regarding these two isolates have been added in the revised manuscript.

6.(Lines 136-137) How did you select these 26 RD-positive supernatants and RD-negative and RT-PCR positive specimens for typing? Why didn’t you analyzed all positive cases?

Response: Samples with Ct values lower than 25 have been selected for typing to allow optimal conditions for getting amplifications with the end-point PCR.

7.(Lines 142-143)   I recommend putting (Table 1) at the end of the sentence “Echoviruses were isolated … (15.8%).

Response: This suggestion has been corrected in the revised manuscript

8.(Line 146) “Tableau” may be French.

Response: This suggestion has been corrected in the revised manuscript

9.(Line 149) A total of 37?

Response: This remark has been corrected in the revised manuscript

  1. (Line 150) 13 should be 18?

 Response: A total of 13 RD-negative and RT-PCR-positive out of 26 had a length of approximately 375 bp and have been used for phylogenetic analyses. This sentence has been rephrased in the revised manuscript for more clarity.

11.(Figure 3) There are several questions related to Figure 3.  

  1. Did you register the sequences you analyzed? If so, please show the GenBank number.

Response: The sequences have been submitted to Genbank and the accession numbers have been added to the revised manuscript (Table S1)

  1. Why you did not include 37 cases shown in Table 1?

Response:  Three out of 36 sequences were too short and were not used for the tree inference.

  1. Information of serotype should be included.

           Response: This remark has been corrected in the revised manuscript

  1. Explanation for brown triangle and green circle should be included.

Response: These explanations have been given the tree’s footnote in the title of the figure 3. The small red triangles indicate the strains from RD-positive supernatants, the green circles indicate the strains from RD-negative and RT-PCR-positive stool samples.

  1. Probably, 19 brown triangles indicate 19 RD-positive supernatants. According to Table 1, there are 3 enterovirus A, 15 enterovirus B, and 1 enterovirus C. However, there are 2 enterovirus A, 17 enterovirus B and 0 enterovirus C in Figure 3.  These data suggest that there is discordance of the results between RIVM program and phylogenetic analysis.    I guess one reason for such a discordance is that partial VP1 sequence were too short (~375bps) and regrettably I think the methodology in this study is not appropriate

Response:  Data analyses were performed again and sequences from the Coxsackievirus A21 (EV-C) as well as one of the Coxsackievirus A16 (EV-A) and  one of the Echovirus 13 (E13) were too short and have been removed from the phylogenetic analysis. The figure 3 has been edited in the revised manuscript.

Reviewer 2 Report

Ndiaye et al. investigated enteroviral etiology for about 300 cases of non-polio acute flaccid paralysis identified in 2017 in various parts of Senegal.  Samples were first investigated by attempting to isolating the virus in cell-culture and Real-Time PCR was performed on negative sample. Overall, enteroviruses were detected in about 30% of the samples. Additionally, from a sub-set of samples, a fragment of the VP1 was obtained for virus typing and this allowed the identifications of viruses within species Enterovirus A, B, and C. Interestingly, there were differences in virus types that could be identified with the 2 methods. Finally, phylogenetic analysis showed that, while all viruses from species C identified in PCR clustered together, in clades A and B both viruses that were positive in cell-culture and in RT-PCR clustered together. Interestingly, Enterovirus B strains formed 2 different clades. The data from this study are highly relevant within the global surveillance of non-polio enterovirus circulation and of acute flaccid paralysis. The study is well performed and presents interesting data, the manuscript is well-written, and conclusions are mostly supported by the results. However, I find some parts of the manuscript a bit confusing, and I think that sequence data could be analyzed in further details. I, therefore, advise the authors to revise the paper before acceptance.

Specific comments:

- Lines 76-85. If for 138 cases 2 samples were collected, the graph shows the number of samples, not the number of notified cases (as stated at lines 84-85). Also, if you had 2 samples for 138 cases and 1 sample for 34, the number of cases would be 172, not 293. Please, clarify this part.  

- Sections 3.1 and 3.2. It would be informative if you could also give the results of the number of patients, besids samples, that tested positive for isolation and/or PCR. Also, were results obtained from the 2 specimens from the same patient concordant?

- Section 3.3. I think it’s worth pointing out that, except from Echo 13, there was no correspondence between viruses that could be successfully isolated since being able to grow in cell culture may be type dependent.

- On the same note, were the Ct values of samples that were PCR-positive but culture-negative high? In other words, did viral load influence the cultivability of the strains?

- Line 149. Why only 32 when you had the sequence of 37 strains? Why did you exclude 5 sequences?

- Phylogenetic analysis. What was the bootstrap value for the C clade? Since it is not indicated I am assuming it is below 80, but it would be handy to know what that value was (79 would be very different from <50). Additionally, one of the 2 B clades includes only sequences from this study. Did you try to blast them to see whether similar sequences were identified before? Additionally, given the position in the tree, these sequences could be recombinant sequences and recombination is something that is frequently reported for enteroviruses. I think it would be worth investigating this hypothesis. Finally, according to your table, you managed to isolate and sequence an Enterovirus C (coxakie A21)… why is this not in the tree? Also, you have only 8 Enterovis C indicated in Table 1, but you have 10 Enterovirus C indicated by a green circle in the tree. How did this happen?

- Lines 185-7 and 229-230. You cannot say this as you did not perform PCR on specimens that were positive in cell-culturing and, therefore, you don’t know how many of those viruses PCR would have picked up.

Minor:

- Line 40. Official taxonomical names of viruses should be written in italics. In this case “Picornaviridae” and “Enterovirus” should be italicized. However, “enteroviruses” is used as common name and it is fine the way it currently is.

- Line 54. When you say “the number of AFP cases has not decreased” do you mean globally or in some specific countries? Also, can you give a temporal indication? Can you provide a reference for this statement?

- Lines 76-81. Please, revise grammar and the use of punctuation in this paragraph.

- Line 118. It should be “midpoint”, not “midpoints”. Also, you used a parechovirus strain to root the tree…

- Line 132. From your methods section, I’d expect that you isolated RNA from 245 stool samples, not 310. Please verify this sentence and clarify it.

- Line 136. How were these 52 samples selected?

- Line 146. There is a typo (Table).

Author Response

Reviewer 2

Open Review

English language and style

( ) Extensive editing of English language and style required
( ) Moderate English changes required
(x) English language and style are fine/minor spell check required
( ) I don't feel qualified to judge about the English language and style

Yes

Can be improved

Must be improved

Not applicable

Does the introduction provide sufficient background and include all relevant references?

(x)

( )

( )

( )

Is the research design appropriate?

(x)

( )

( )

( )

Are the methods adequately described?

( )

(x)

( )

( )

Are the results clearly presented?

( )

( )

(x)

( )

Are the conclusions supported by the results?

( )

(x)

( )

( )

Comments and Suggestions for Authors

Ndiaye et al. investigated enteroviral etiology for about 300 cases of non-polio acute flaccid paralysis identified in 2017 in various parts of Senegal.  Samples were first investigated by attempting to isolating the virus in cell-culture and Real-Time PCR was performed on negative sample. Overall, enteroviruses were detected in about 30% of the samples. Additionally, from a sub-set of samples, a fragment of the VP1 was obtained for virus typing and this allowed the identifications of viruses within species Enterovirus A, B, and C. Interestingly, there were differences in virus types that could be identified with the 2 methods. Finally, phylogenetic analysis showed that, while all viruses from species C identified in PCR clustered together, in clades A and B both viruses that were positive in cell-culture and in RT-PCR clustered together. Interestingly, Enterovirus B strains formed 2 different clades. The data from this study are highly relevant within the global surveillance of non-polio enterovirus circulation and of acute flaccid paralysis. The study is well performed and presents interesting data, the manuscript is well-written, and conclusions are mostly supported by the results. However, I find some parts of the manuscript a bit confusing, and I think that sequence data could be analyzed in further details. I, therefore, advise the authors to revise the paper before acceptance.

Specific comments:

- Lines 76-85. If for 138 cases 2 samples were collected, the graph shows the number of samples, not the number of notified cases (as stated at lines 84-85). Also, if you had 2 samples for 138 cases and 1 sample for 34, the number of cases would be 172, not 293. Please, clarify this part.  

Response: We thank the reviewer for this comment. The number of cases has been corrected in the revised manuscript.

- Sections 3.1 and 3.2. It would be informative if you could also give the results of the number of patients, besids samples, that tested positive for isolation and/or PCR. Also, were results obtained from the 2 specimens from the same patient concordant?

Response: Details regarding the number of patients from whom the stool samples tested positive has been added in the revised manuscript. Two distinct positive specimens from the same patient haven’t been found in our study.

- Section 3.3. I think it’s worth pointing out that, except from Echo 13, there was no correspondence between viruses that could be successfully isolated since being able to grow in cell culture may be type dependent.

Response: More details have been added to the section 3.1.3 and the discussion in the revised manuscript

- On the same note, were the Ct values of samples that were PCR-positive but culture-negative high? In other words, did viral load influence the cultivability of the strains?

Response: We have found that the viral load influences the cultivability of the strains belonging to EV-A and EV-B species while no effect have been observed for the EV-C species.

- Line 149. Why only 32 when you had the sequence of 37 strains? Why did you exclude 5 sequences?

Response: This section has been corrected in the revised manuscript. Finally, 3 out of 36 sequences have been excluded for the phylogenetic analysis because they were too short. Details have been added in the revised manuscript.

- Phylogenetic analysis. What was the bootstrap value for the C clade? Since it is not indicated I am assuming it is below 80, but it would be handy to know what that value was (79 would be very different from <50). Additionally, one of the 2 B clades includes only sequences from this study. Did you try to blast them to see whether similar sequences were identified before? Additionally, given the position in the tree, these sequences could be recombinant sequences and recombination is something that is frequently reported for enteroviruses. I think it would be worth investigating this hypothesis. Finally, according to your table, you managed to isolate and sequence an Enterovirus C (coxakie A21)… why is this not in the tree? Also, you have only 8 Enterovis C indicated in Table 1, but you have 10 Enterovirus C indicated by a green circle in the tree. How did this happen?

Response: We thank the reviewer for these comments. The ML tree and the Table 1 have been amended in the revised manuscript. Sequences from the strains in the B3 clade including from Senegal have been analyzed for recombination but no evidence of recombination have been found. However, further characterizations based on whole genome sequencing will be performed in future studies. Three out of 36 sequences including the Coxsackievirus A21 have been excluded for the phylogenetic analysis because they were too short.

- Lines 185-7 and 229-230. You cannot say this as you did not perform PCR on specimens that were positive in cell-culturing and, therefore, you don’t know how many of those viruses PCR would have picked up.

Response: RT-PCR has been performed on all the 310 stool samples cited in section 3.1.2.

Minor:

- Line 40. Official taxonomical names of viruses should be written in italics. In this case “Picornaviridae” and “Enterovirus” should be italicized. However, “enteroviruses” is used as common name and it is fine the way it currently is.

Response: These remarks have been corrected in the revised manuscript

- Line 54. When you say “the number of AFP cases has not decreased” do you mean globally or in some specific countries? Also, can you give a temporal indication? Can you provide a reference for this statement?

Response: This sentence has been rephrased in the revised manuscript for more clarity.

- Lines 76-81. Please, revise grammar and the use of punctuation in this paragraph.

Response: This remark have been corrected in the revised manuscript

- Line 118. It should be “midpoint”, not “midpoints”. Also, you used a parechovirus strain to root the tree…

Response: This typo has been corrected in the revised manuscript. We do not understand the point of the reviewer regarding the used outgroup.

- Line 132. From your methods section, I’d expect that you isolated RNA from 245 stool samples, not 310. Please verify this sentence and clarify it.

Response: RT-PCR was performed on all the 310 stool samples.

- Line 136. How were these 52 samples selected?

Response: Samples with Ct values lower than 25 have been selected for typing to allow optimal conditions for getting amplifications with the end-point PCR.

- Line 146. There is a typo (Table).

Response: This typo has been corrected in the revised manuscript

Reviewer 3 Report

NPEVs have been frequently reported to be in association with AFP cases worldwide and  they are well known to be responsible for many other illnesses such as self-limiting febrile illnesses, meningitis, encephalitis, myocarditis, pancreatitis and foot and mouth disease.  An efficient NPEVs laboratory diagnosis together with the development  of national NPEV surveillance networks are essential to provide better estimation on NPEVs disease burden. An Algorithm for laboratory diagnosis  must  consider the broad spectrum of clinical manifestations.

This study could provide a message for public health institutions in Senegal and could be useful for a future development of a national surveillance network for NPEVs circulation.

However, the manuscript of Ndack Ndiaye et al. has several limits to obtain partial, but significant, information of NPEVs circulation in Senegal. Indeed, the analysis was conducted in short time period, only AFP cases were considered of entire broad spectrum of NPEVs clinical manifestations and few NPEVs were molecular typed. Moreover, the aims of the study are not very clear and the title of the paper is not appropriate. Figures 1 and 2 contain data that seem to be not very relevant and the usefulness of the phylogenetic tree is not clear. Indeed, the information provided by the phylogenetic analysis is already contained in Table 1. 

I suggest reviewing  the manuscript and presenting  it as a Brief Communication.

Author Response

Reviewer 3 :

Open Review

English language and style

( ) Extensive editing of English language and style required
( ) Moderate English changes required
( ) English language and style are fine/minor spell check required
(x) I don't feel qualified to judge about the English language and style

Yes

Can be improved

Must be improved

Not applicable

Does the introduction provide sufficient background and include all relevant references?

( )

(x)

( )

( )

Is the research design appropriate?

( )

( )

(x)

( )

Are the methods adequately described?

( )

(x)

( )

( )

Are the results clearly presented?

( )

( )

(x)

( )

Are the conclusions supported by the results?

( )

( )

(x)

( )

Comments and Suggestions for Authors

NPEVs have been frequently reported to be in association with AFP cases worldwide and  they are well known to be responsible for many other illnesses such as self-limiting febrile illnesses, meningitis, encephalitis, myocarditis, pancreatitis and foot and mouth disease.  An efficient NPEVs laboratory diagnosis together with the development  of national NPEV surveillance networks are essential to provide better estimation on NPEVs disease burden. An Algorithm for laboratory diagnosis  must  consider the broad spectrum of clinical manifestations.

This study could provide a message for public health institutions in Senegal and could be useful for a future development of a national surveillance network for NPEVs circulation.

However, the manuscript of Ndack Ndiaye et al. has several limits to obtain partial, but significant, information of NPEVs circulation in Senegal. Indeed, the analysis was conducted in short time period, only AFP cases were considered of entire broad spectrum of NPEVs clinical manifestations and few NPEVs were molecular typed. Moreover, the aims of the study are not very clear and the title of the paper is not appropriate. Figures 1 and 2 contain data that seem to be not very relevant and the usefulness of the phylogenetic tree is not clear. Indeed, the information provided by the phylogenetic analysis is already contained in Table 1.  

I suggest reviewing the manuscript and presenting it as a Brief Communication.

Response: We thank the reviewer for these comments. A total of 52 NPEVs were molecular typed and samples with Ct values lower than 25 have been selected for more efficient amplification since end-point PCR could be at all less sensitive than RT-qPCR. The objectives have been rephrased in the revised manuscript for more clarity. Details regarding Figures 1 and 2 and the importance of the phylogenetic tree have been also added to the revised manuscript.

Round 2

Reviewer 2 Report

This version of the manuscript is significantly improved with respect to the previous one and the study design is much clearer. However, there are still a few small things that need to be clarified: 

- Line 25. I would mention here that 89 samples were positive in PCR, rather than stating how many were negative in cell culture

- Line 112. Shouldn't this be from ALL samples?

- Lines 134 and 232. Replace "and Maximum Likelihood (ML) methods, respectively," with "method". 

- Line 142. The tree was not rooted on midpoint, but with an outgroup (parechovirus). Therefore, you can just say that a parechovirus strain was used as an outgroup. 

- Section 3.1.2. Did I understand correctly that, overall, 89/172 (51.7%) cases of AFP were NPEVs? This is a result worth mentioning here. 

- Lines 200-206. I would be more clear here in stating that these results were obtained from samples and not from culture supernatant. Maybe you can write "stool samples" at lines 202 and 203...

- Line 257. You should specify here that these values are detection rates and not sensitivity values. 

Reviewer 3 Report

Although authors have modified the manuscript, several limits remain. The weakness in this manuscript lies lacks of clarity of the aim of the study. Is not clear if the main goal of the work is to provide new insights in the epidemiology of NPEVs or the comparison between cell culture and Real-Time methods. However, for both goals, there are many points that should be reviewed. Indeed, the analysis was conducted in short time period, only AFP cases were considered of entire broad spectrum of NPEVs clinical manifestations and few NPEVs were molecularly typed. Moreover, the study lacks of the correct methodological approach to “ ... find the most sensitive method  for NPEVs detection …” (Line 71). 

Lastly, the doubts expressed in the previous comment, on the usefulness of the figures and the phylogenetic tree, remain.

For these reasons, the article should be rejected.